# Oral Nutritional Supplementation Affects the Dietary Intake and Body Weight of Head and Neck Cancer Patients during (Chemo) Radiotherapy

**DOI:** 10.3390/nu12092516

**Published:** 2020-08-20

**Authors:** Isabela Borges Ferreira, Emanuelle do Nascimento Santos Lima, Paula Philbert Lajolo Canto, Cristiana Araújo Gontijo, Yara Cristina de Paiva Maia, Geórgia das Graças Pena

**Affiliations:** 1Graduate Program in Health Sciences, School of Medicine, Federal University of Uberlandia, 1720 Para Avenue, 2H, Uberlandia MG 38400-902, Brazil; isaborgesferreira@gmail.com (I.B.F.); emanuellensl@yahoo.com.br (E.d.N.S.L.); cristianaagontijo@hotmail.com (C.A.G.); yara.maia@ufu.br (Y.C.d.P.M.); 2Department of Oncology, Clinical Hospital of Federal University of Uberlandia, 1888 Para Avenue, Uberlandia MG 38405-320, Brazil; paula.lajolo@oncocentro.com.br

**Keywords:** food consumption, weight loss, malnutrition, dietary supplements, head and neck neoplasms

## Abstract

Considering the symptoms of (chemo) radiotherapy and the reduction in food intake in head and neck cancer (HNC) patients, this study aimed to investigate the association between treatment time points and oral nutritional supplementation (ONS) on dietary intake to estimate the frequency of energy and nutrient inadequacy, and also to evaluate body weight changes (BWC). Dietary intake data of 65 patients were obtained from 24-h dietary recalls and prevalence of inadequacy was calculated before or at the beginning (T0), in the middle (T1), and at the end of treatment (T2). BWC were calculated as the weight difference considering the previous weight reported and/or measured. Energy and macronutrient intake decreased in T1 and then improved in T2 (*p* < 0.001 for both). Micronutrient intake increased during treatment due to ONS use, but still presented a high probability of inadequate intake. In particular, calcium, magnesium, and vitamin B6 showed almost 100% of probability of inadequacy for those who did not use ONS. Finally, overweight patients suffered a higher weight accumulated deficit with a delta of −15 kg compared to other BMI (body mass index) categories. Therefore, we strongly recommend initiating nutritional counseling in conjunction with prophylactic ONS prescription from diagnosis to adjust nutrient intake and minimize weight loss.

## 1. Introduction

Head and neck cancer (HNC) are the seventh most common malignant tumors in the world [1]. Among the modalities of treatment, chemoradiotherapy (CRT) is related to symptoms such as oral mucositis, xerostomia, and dysgeusia, which could affect dietary intake [2,3]. Aside from symptoms resulting from treatment, these patients also have other negative symptoms as the tumor can cause problems when chewing and swallowing [4], and make eating difficult and painful [5].

These symptoms can cause a reduction in dietary intake [5], and are associated with weight loss and malnutrition [6,7], worse quality of life [4], infection, higher hospital readmissions, longer length of hospital stay, and mortality [8]. As a result, a marked change in the consistency of food consumption by HNC patients has been observed during the treatment period that may interfere with energy adequacy [5,9] and lead to reduced macronutrient intake. Furthermore, inadequate intake of micronutrients such as vitamins D, E, C, folate, calcium, iron, and magnesium has been described in HNC patients, and oral nutritional supplementation (ONS) has been required in order to achieve recommended levels [10,11], since micronutrients are important for enzymatic reactions that impact the metabolism as a whole [12].

According to the United Kingdom National Multidisciplinary Guidelines [13], nutritional support is an important part of the treatment of HNC patients. Nutritional intervention during treatment is indicated in order to prevent weight loss, increase food intake, and decrease treatment interruptions. Thus, when oral nutrition is inadequate, ONS use and tube feeding are indicated. HNC patients who receive nutritional counseling and use nutritional support show improvement in weight loss, quality of life and survival [14,15,16,17].

Thus, nutritional support of HNC patients including ONS use is important. However, the levels of both dietary intake and ONS contribution to macro- and micronutrient consumption during treatment are currently unknown or understudied. Studies in the literature assess the presence of individual nutritional counseling [18] and use or not of ONS [19] with outcomes such as weight [15,20], quality of life, mortality, and nutritional status [21]. However, as these variables are abstract, studies that account for or assist in quantifying the nutrient intake are necessary, since the quantification of consumption is often neglected in clinical practice and little explored in studies due to the complexity of analysis. Thus, with this evaluation, it is possible to obtain an assertive nutritional approach. Moreover, most studies evaluating dietary intake in HNC patients are long-term [10,22], and do not assess the effects and acute treatment changes presented by these patients.

To the best of our knowledge, this is the first prospective study to evaluate macro and micronutrient intake during treatment; that is, in the short-term. This is important because it allows dietary nutritional monitoring to be frequently carried out in order to minimize the possible cumulative losses that the patient has since the onset of symptoms and which are often not properly valued in these studies. In addition, there are no data in the literature on the prevalence of inadequate dietary intake in these patients. It is therefore important to identify the treatment time point at which the impact on dietary intake is greatest, so that an early assessment of nutritional changes can be made and the above-mentioned negative impacts minimized through nutritional counseling. Furthermore, it is possible to minimize weight loss by considering not only nutritional counseling, but the identification of specific needs, taking care to not neglect patients who do not demonstrate physical malnutrition so clearly.

We hypothesized that macro- and micronutrient intake was reduced in HNC patients during treatment. Thus, the aim of this prospective study was to investigate the association between treatment time points and ONS on dietary intake to estimate the frequency of energy and nutrient inadequacy as well as to evaluate body weight changes (BWC) in HNC patients during (chemo) radiotherapy.

## 2. Materials and Methods

### 2.1. Study Design and Ethical Aspects

A prospective observational study was carried out with HNC patients during (chemo) radiotherapy. These patients were recruited from the outpatient treatment of a tertiary university hospital, which was the regional referral center for HNC patients undergoing antineoplasic treatment in the city of Uberlandia, Minas Gerais, Brazil, between July 2017 and November 2018. The patients were evaluated at three time points: before or at the beginning of treatment (T0); in the middle (T1 ~four weeks,) and at the final treatment (T2 ~eight weeks).

This study was approved by the Human Research Ethics Committee (protocol number 65340116.8.0000.5152) and all participants signed a free and informed consent form. The entire study was conducted based on the standards of the Declaration of Helsinki [23].

### 2.2. Sample Size and Eligibility Criteria

In order to estimate the sample size required for this study, we used G* Power software, version 3.1 (Heinrich-Heine-University Düsseldorf, Düsseldorf, Germany). Considering a single group of individuals and three measurements, the sample size calculations were based on an F-test repeated-measures ANOVA with an effect size of 0.25, an α level of 0.05, and 95% power. The result of the calculation required a minimum sample size of forty-five patients. So, considering a 20% adjustment for possible losses, a minimum of fifty-four patients was needed at baseline (T0).

The inclusion criteria were patients aged 18 years or older, diagnosed with primary malignant tumors in the head and neck region who were undergoing radiotherapy (RT), chemotherapy (CT), or a combination of these modalities, with or without surgery, independent of tumor stage, and were invited to participate. Patients were considered to be T0 when treatment had not been initiated or at the beginning of treatment, before presenting collateral effects. Patients with metastasis at T0 or who had been previously treated with RT and/or CT for other types of cancer in the last 10 years were excluded.

During the study period, 140 patients were approached, 25 declined to participate, and 24 did not meet the inclusion criteria, resulting in 91 patients (60 patients with complete dietary intake data in T0). Of the 91, two died, 16 lost follow-up, and three refused to continue in the study, totaling 70 patients in T1 (56 patients with complete dietary intake data in T1). Considering the total participants in T1, one died and six lost follow-up, totaling 63 patients in T2 (53 patients with complete dietary intake data in T2). With respect to patients with complete intake data, five patients were included only at T1 and T2 and nine patients were included only at T0 and T2, resulting in 65 patients with a complete dietary assessment.

### 2.3. Demographic, Clinical, and Nutritional Assessment

Characteristics such as age (years), sex (female/male), clinical diagnosis, tumor site, tumor stage by American Joint Committee on Cancer–AJCC [24], treatment schedule, ONS or gastric or enteral tubes use and nutritional counseling were obtained from an initial structured questionnaire and medical records.

Height and body weight were measured at three study time points using standard protocols [25] and weight at six (habitual weight) and one month ago was self-reported. Body Mass Index (BMI) was calculated as body weight (kg)/height (m²) for assessment and classification of nutritional status. The patients were classified into three groups based on the World Health Organization criteria of underweight (<18.5 kg/m^2^), normal weight (18.5–24.9 kg/m^2^), and overweight (≥25.0 kg/m^2^) [25].

BWC were calculated as the weight difference considering the weight for six months (habitual weight) and one month self-reported before treatment, and measured at the three time points of the study (T0, T1, and T2). BWC calculation considering five points (two points before treatment and the three treatment time points): BWC T_6mo_ = 0; BWC T_1mo_ = weight for 1 month – habitual weight; BWC T0 = weight T0 – habitual weight; BWC T1 = weight T1 – habitual weight; BWC T2 = weight T2 – habitual weight. To calculate the habitual BMI, weight for six months (habitual weight) was used and height was measured at T0. BWC calculation considering three points (the three treatment time points): BWC T0 = 0; BWC T1 = weight T1 – weight T0; BWC T2 = weight T2 – weight T0. To calculate the BMI at T0, the weight and height measured at T0 were used.

### 2.4. Dietary Assessment

Dietary and ONS intake data were obtained from 24-h dietary recalls (24HR) applied at three study time point (T0, T1, and T2). At each time point, the first 24HR was applied in person and other two by telephone interview on non-consecutive days including weekend days, in order to record the eating habits of the study participants more precisely, ideally totaling nine 24HRs for each patient.

Data were collected using the United States Department of Agriculture Multiple-Pass Method, which guides the respondent to respond to 24HR in five steps [26]. ONS intake was calculated using the manufacturers’ labels in order to include all nutrients coming from this source. The same process was used for patients who were given nutritional therapy by tube feeding. Since the patients can use different brands of the ONS, depending on the chosen manufacturer, the quantity and brand labels were properly registered in order to provide complete accounting of nutrient supply.

The following nutrients were analyzed using 24HR: total energy (kcal), carbohydrate (g), protein (g), lipid (g), dietary fiber (g), monounsaturated, polyunsaturated and saturated fats (g), total cholesterol (mg), thiamine (mg), riboflavin (mg), niacin (mg), vitamin B6 (mg), vitamin C (mg), iron (mg), magnesium (mg), zinc (mg), calcium (mg), phosphorus (mg), manganese (mg), potassium (mg), and sodium (mg). Nutrient content was estimated by Dietpro software, version 5.8.1 (Dietpro Viçosa, MG, BR), using, for preference, the Brazilian Table of Food Composition [27]. For foods not found in this table, the United States Department of Agriculture [28] table was used. Nutritional content from food or supplements not found in the software’s tables were added based on their labels.

Energy and nutrient values were adjusted for intra-individual variability due to intrinsic dietary intake variability, in order to obtain an individual consumption estimate of energy and nutrients using the Personal Computer version of the Software for Intake Distribution Estimation (PC-SIDE) (Department of Statistics, Iowa State University, Iowa, USA), following the methodology described by Nusser et al. [29]. Subsequently, these were adjusted by the residual method for the sample total energy in order to adjust nutrient estimates [30].

Prevalence of inadequacy was calculated by the Estimated Average Requirement (EAR) method as a cut-off point [31]. The Z value was calculated ((EAR—average intake)/standard deviation)) and the Z table curve was consulted to verify the corresponding percentage of individuals with intakes below EAR. For this evaluation, the values of energy and nutrients adjusted only for intra-individual variability were used.

For fiber, manganese, potassium, and sodium, for which there are no established EAR values, an intake comparison was made with their respective adequate intake (AI) values. When these nutrients showed an intake above AI, adequacy regarding tolerable upper intake level (UL) was verified. Macronutrient distribution to total energy value was analyzed using acceptable macronutrients distribution range (AMDR) values as a reference [32].

For energy (25 kcal/kg/day) and protein intake (1 g/kg/day), the European Society for Clinical Nutrition and Metabolism (ESPEN) recommendations were used as reference values [33] to understand whether individuals are capable of achieving the recommended minimum levels of intake of these macronutrients. For cholesterol and monounsaturated, polyunsaturated, and saturated fats, we used the recommendations of the Food and Agriculture Organization of the United Nations [34].

### 2.5. Statistical Analyses

Variable distributions were evaluated by the Kolmogorov–Smirnov test. Descriptive statistics were shown in percentage, mean, and standard deviation to describe the characteristics of the investigated population.

Generalized estimating equations (GEE) models were used to determine the association of treatment time points (T0, T1, and T2), ONS use and treatment time points, and ONS interactions (independent variables) with nutrient consumption (dependent variables). GEE is a method that considers the association between different observations in the same individual in prospective studies, performing a better evaluation of repeated-measures data [35]. The gamma, linear, or Tweedie distribution models were individually tested for all outcomes. Lower quasi-likelihood under the independence model criterion (QIC) was observed in the gamma with the log-link model, and was chosen for GEE analysis. The Bonferroni post-hoc test was used to adjust for multiple comparisons. Type of treatment, sex, age (years), tumor site, and stage were considered as confounders.

ONS use was grouped in order to be evaluated as an exposure in the GEE models. Since the ONS can be indicated at the beginning, middle, or final treatment, the individuals were categorized according the frequency of ONS use by time points: individuals who used ONS 2/3 times (higher frequency of use) and those who used ONS 0/1 time (no or lower frequency of use). This strategy was also used to fix this exposure to analyze the longitudinal effect of ONS use on the dietary intake. For statistical tests not performed by GEE, the individuals were categorized into “with ONS” or “without ONS” at each treatment time point. Confidence interval (CI) of 95% and *p*-value < 0.05 were considered as levels of statistical significance. All data were analyzed using Statistical Package for Social Sciences (SPSS), version 25.0 (SPSS Inc., Chicago, IL, USA).

## 3. Results

Of the 65 patients, the mean age was 59.8 ± 10.1 years, 53 (81.5%) patients were male, 26 (40.0%) had a tumor in the larynx, followed by oral cavity 21 (32.3%), and pharynx 14 (21.5%). The majority of patients were at an advanced (T3–T4; 58.4%) stage of cancer. Among the treatment types, most prevalent were CRT 33 (50.8%) and RT 16 (24.6%) (Table 1). Antineoplasic treatment lasted approximately eight weeks with a daily RT session from Monday to Friday, totaling 38 to 40 sessions. According to the institutional protocol, patients underwent RT with a total final radiation dose of 70 or 72 Gy with daily doses of 180 or 200 cGy. Mean ± standard deviation of the number of RT sessions performed by the study patients was 1.58 ± 2.3 at time T0, 20.27 ± 3.7 at time T1, and 36.27 ± 3.7 at time T2. For patients undergoing CRT, the protocol consisted of weekly cisplatin during the radiotherapy course.

Regarding nutritional status performed by BMI, the frequency of underweight increased in the middle 14 (23.3%), and at the end of treatment 13 (22.0%), and overweight decreased in the middle 16 (26.7%) and at the end of treatment 13 (22.0%), both compared to before treatment. ONS use increased during treatment (Table 1).

The association of treatment time points and group of ONS use on energy, macro-, and micronutrient intake is shown in Table 2. In general, there was a significant reduction in energy (*p* < 0.001), macronutrients (*p* < 0.001), and cholesterol (*p* < 0.001) intake from the beginning (T0) to the middle of treatment (T1), with the increase at the end of treatment (T2). In addition, those who used ONS 2/3 times consumed more protein and less polyunsaturated fat (1.32 g/kg/day and 16.95 g, respectively) than those who used ONS 0/1 time (1.18 g/kg/day and 19.32 g, respectively) at all treatment time points, with a difference of +0.14 g/kg/day for protein (*p* = 0.031) and −2.37 g of polyunsaturated fat (*p* = 0.005). For micronutrients, except for niacin and vitamin C, an increase in intake from the beginning (T0) to the middle of treatment (T1) was observed and the values were maintained at the end of treatment (T2). The use of ONS 2/3 times increased intake of all micronutrients, except potassium and sodium (Table 2). 

Percentage and prevalence of energy and nutrient inadequacy is shown in Table 3 for macronutrients and Table 4 for micronutrients. In general, a high percentage and prevalence of inadequacy was observed for energy and nutrient intake, especially in patients who did not use ONS. Macronutrients with the highest percentage of inadequacy according to AMDR were carbohydrates, followed by lipids, monounsaturated fats, and saturated fats. Moreover, even using ONS at T1, protein (grams), lipids, monounsaturated, polyunsaturated, and saturated fat intake presented a high percentage of inadequacy.

In Table 4, we observed that even though there was an increase in micronutrient intake during treatment with ONS use (Table 2), it was not enough to ensure adequacy compared to EAR. Inadequacy prevalence was lower in those who used ONS compared with those who did not. However, a high prevalence of inadequacy was observed, mainly for calcium, magnesium, and vitamin B6 intake, which was almost 100% for those who did not use ONS. Compared with AI, male patients presented a higher percentage of values below the recommended levels for fiber and manganese intake.

Regarding the initial nutritional status (six months before the treatment), the patients suffered negative mean BWC, and the overweight patients suffered a higher weight accumulated deficit with a delta of −15 kg (Figure 1A). Furthermore, those who used ONS showed less weight loss, except of the overweight patients (Figure 1B).

There was a decreasing trend of energy (kcal/kg/day, Figure 2A–C) and protein intake (g/kg/day, Figure 2D–F) comparing the underweight with overweight patients regardless of ONS use. Only the underweight group with and without ONS use showed an important difference considering the energy in T1 and protein consumption in T2 (Figure 2B,F).

## 4. Discussion

A significant reduction in energy, macronutrients, and cholesterol intake at the middle of treatment (first month) and the return of these consumption levels at the end of treatment (second month) were observed. Regarding micronutrients, the majority of patients increased their intake from the beginning to the middle of treatment and the values were maintained until the end due to ONS use. However, energy, macro-, and micronutrient inadequacy prevalence was high in all time points, especially in patients who did not use ONS. Finally, overweight patients suffered a higher weight accumulated deficit compared to other BMI categories. In general, those who used ONS showed less weight loss, except for overweight patients, and only the underweight group with and without ONS use showed an important difference considering the energy in T1 and protein consumption in T2.

Recent studies in the literature have evaluated dietary intake after treatment (long-term), but not during the treatment. They assessed at diagnosis and post induction chemotherapy, after RT, from one and three months after the end of treatment [22], or at baseline and post-treatment (after 4–6 weeks of RT and/or CT, and follow-up (8–10 weeks after completion of treatment) [10]. The difference in our study is that we identified the treatment time points at which there was the greatest impact on dietary intake and showed the high prevalence of the inadequacy of energy and nutrients (short-term). This is important because the greater impact during treatment can cause negative consequences that can be predicted if the patient is monitored, even if the decrease in food intake and weight values is expected. In addition, we used GEE modeling to estimate not only the effects of treatment time points on dietary intake, but also the contribution of ONS on nutrient intake. Besides the lack of studies on dietary intake and HNC in the literature, there is also no data on the prevalence of inadequacy. These data are important to assess whether the amounts of micronutrients ingested are sufficient. This can contribute to an adequate nutritional approach and may reduce the risk of adverse health outcomes in these patients, since, as evidenced in this study, even having increased the intake of micronutrients with ONS use, patients consumed micronutrients in quantities below the recommendation. Furthermore, to the best of our knowledge, no study has adjusted for intra-individual variability of values related to energy and nutrient consumption in order to better estimate individual intake, as suggested by Nusser et al. [29]; nor have they adjusted for the total energy of the sample by the residual method due to the association between energy and most nutrients, as recommended in the literature [30]. Thus, the studies that evaluated the food intake of patients with HNC have not performed these necessary analyses.

As symptoms presented by HNC patients such as xerostomia, dysgeusia, dysphagia, mucositis, and thick saliva [36,37] can limit oral intake [22] and lead to changes in food consistency, our results for macronutrients were expected. A lower macronutrient intake is expected when patients opt for pasty, liquid, or mild food [9], impacting in particular fiber content. Additionally, an increase in intake of soup and foods prepared with milk has been reported [10,36]. The same decreased intake seem in studies that assessed the dietary intake in the long-term (at diagnosis, post treatment, and follow-up) have found a reduction in energy and protein intake in HNC patients [7,22]. Adequate protein intake can minimize the severity of oral mucositis in patients with HNC undergoing RT due to the ability of protein to maintain integrity or repair mucosal lesions [38]. On the other hand, a low protein intake can increase the risk of fatigue and mortality in advanced cancer patients undergoing CT [39].

Cancer patients present impaired macronutrient metabolism due to systemic inflammation, which can lead to altered protein turnover, loss of fat and muscle mass, increased production of acute phase proteins, insulin resistance, glucose intolerance, and increase or maintenance of lipid oxidation capacity [33]. Therefore, impairment of these metabolic pathways has a negative effect on clinical outcome and macronutrients are needed for bodily maintenance and better response to treatment. However, in this study, although patients who used ONS consumed more protein than those who did not, protein intake in grams presented a higher percentage of inadequacy at T1. Therefore, nutritional intervention with complete assessment is extremely important to make changes in the diet of these patients, avoiding recommendations directed only at increasing energy and protein intake without assessing food quality. Furthermore, it is also important to assess the amount ingested and not only the use or not of ONS, or indicate its use without providing guidance to patients, because, depending on the severity and if the ONS does not have the amount of proteins necessary to minimize the impacts, protein modules can be used in order to achieve nutritional requirements as soon as possible.

Unlike macronutrients, the mean intake of micronutrients increased over the duration of treatment. This suggests that ONS may have been determinant in increasing micronutrient intake, although not at a high enough level to ensure adequacy. A high prevalence of energy and nutrient inadequacy was observed despite ONS use, mainly for calcium, magnesium, and vitamin B6. This reveals that HNC patients require nutritional intervention with a special attention to the quantification of food intake in order to estimate possible deficits and achieve adequate levels of macro- and micronutrients through prophylactic ONS prescription. Previous studies have highlighted the importance of vitamin B6 as a protective factor against the development of cancer [40], antioxidant effects [41], and increased immune response [42].

In addition, magnesium participates in energy metabolism, protein synthesis, and plays an important physiological role in organs such as the brain and heart [43]. In relation to low micronutrient intake, there is a high prevalence of vitamin D deficiency among HNC patients [44]. This deficiency has been linked to an increased risk of postoperative hypocalcemia in patients undergoing total thyroidectomy [45]. Low levels of calcium, vitamin E, and folate intake were also found in patients with HNC [10,44]. So, while adequate levels of some micronutrients can be obtained from a healthy diet, inadequate dietary intake of others can lead to negative health consequences.

Moreover, our study also showed an increase in ONS usage by time points. In other words, ONS complements other food intake, being an alternative route to achieving the recommended levels of micronutrient intake [10]. Recent studies have shown less CRT-related toxicity, better weight maintenance, and tolerance to treatment with nutritional counseling using ONS [15,19,46].

Although micronutrient intake and ONS use increased during treatment, the number of malnourished patients also increased and overweight patients decreased between time points. Thus, a decrease in BMI and an increase in malnutrition led to more ONS prescription and to more frequent use, since these patients do not use ONS prophylactically. This situation may be due to late diagnosis or advanced age as well as low dietary intake and treatment side effects.

According to the ESPEN guidelines, nutritional intervention including ONS provision is recommended to ensure adequate dietary intake, prevent weight loss, and avoid treatment discontinuation in HNC patients [33]. However, as previously mentioned, the majority of patients did not receive nutritional counseling at the beginning of treatment and approximately one third did not receive it at any time point. Thus, nutritional interventions through individualized dietary counseling can positively influence long-term outcomes related to quality of life and nutritional status [14,21,47].

Similar to other studies [15,16,19], we observed that those who used ONS showed less weight loss (except of the overweight) and consumed more protein (0.14 g/kg/day). Additionally, malnourished patients lost less weight, while overweight patients lost more weight, showing that the initial BMI defines the BWC. An additional finding was an accumulated deficit of weight loss that was found before the start of treatment. Corroborating these findings, Orell et al. [16] found that overweight patients lost more weight due to symptoms such as anorexia and nausea. According to the Aspen Guideline [48], critically ill patients with obesity have greater complications when compared to patients with normal weight classified by BMI and have an indication for assessment and early nutritional support. These results are important because they show that there is a significant cumulative weight loss before treatment and that overweight patients may lose more weight not only due to the presence of the disease and the symptoms presented, but also because they are not receiving nutritional counseling according to their needs. Therefore, more attention should be given to overweight patients, since they are often neglected and have been little explored by studies, despite presenting more mass and worse deficit.

Thus, this study showed how much macronutrient intake decreased during treatment and that, even consuming ONS, micronutrient intake is below that recommended. Furthermore, it also demonstrated that the patients suffered negative mean BWC and the overweight patients had the highest accumulated weight deficit. This reinforces the importance of following the Guidelines for HNC patients [13] in which they guide the weekly consultation with a dietitian during treatment to obtain better results. Therefore, we strongly recommend that dietary counseling consultations are routinely provided following cancer diagnosis in order to carry out more specific nutritional orientations such as food recipes fortified with ONS, which will improve caloric intake and maintain adequate intake, avoiding a deficit in nutrient consumption that will cause weight loss and increases in malnutrition, thereby helping the recovery of HNC patients.

This study has some limitations. 24HR was used, and although it is the most accurate tool for dietary intake analysis, it may present a memory bias because it depends on an individual’s ability to accurately recall their food intake. However, in order to minimize this limitation, interviews were conducted by trained dietitians and there was standardization at the time point of collection of 24HR as well as the use of the multiple pass method and typing in order to obtain more reliable results. Moreover, nutrient intake was adjusted for intra-individual variability and for energy intake in order to present intake estimates as precisely as possible. Finally, we did not separate ONS from enteral nutrition because there was no difference in consumption. Therefore, what was consumed in both systems was evaluated.

The study also has strengths. It evaluates the short-term impact on dietary intake, which is during treatment, and assesses the prevalence of inadequate micronutrient intake. This is clinically important since we can minimize negative health outcomes such as malnutrition, delay in post-treatment recovery, longer convalescence, and other long-term impacts such as decreased quality of life and mortality.

## 5. Conclusions

Head and neck cancer patients showed energy and macronutrient intake decreased at the middle of treatment and the increased micronutrient intake due to ONS use. Despite this, the prevalence of inadequate energy and nutrient intake, particularly for calcium, magnesium, and vitamin B6 was high in all time points even with ONS use, but proved worse for those who did not use ONS. Furthermore, overweight patients suffered a higher weight accumulated deficit compared to other BMI categories. Patients on ONS showed a lower weight deficit. Therefore, we strongly recommend initiating nutritional counseling from diagnosis to optimize macronutrient intake in conjunction with prophylactic ONS prescription to adjust micronutrient intake and minimize the weight loss, making it possible to prevent worse prognosis and nutritional status.

## Figures and Tables

**Figure 1 nutrients-12-02516-f001:**
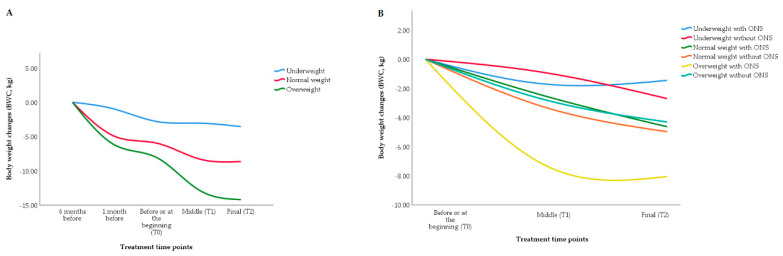
BWC by BMI groups. (**A**) BWC considering 5 time points (two points before treatment and the three treatment time points) for the patients categorized by habitual BMI (kg/m²). (**B**) BWC for the patients considering three time points (the three treatment time points) categorized by T0 BMI (kg/m²) and use or not of ONS. Abbreviations: BWC, body weight changes; BMI, body mass index; ONS, oral nutritional supplementation.

**Figure 2 nutrients-12-02516-f002:**
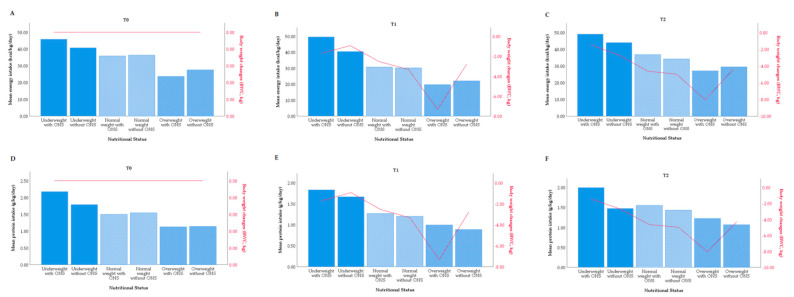
Mean energy and protein intake and BWC by nutritional status. (**A**) Mean energy intake and BWC considering three time points by nutritional status in T0. (**B**) Mean energy intake and BWC considering three time points by nutritional status in T1. (**C**) Mean energy intake and BWC considering three time points by nutritional status in T2. (**D**) Mean protein intake and BWC considering three time points by nutritional status in T0. (**E**) Mean protein intake and BWC considering three time points by nutritional status in T1. (**F**) Mean protein intake and BWC considering three time points by nutritional status in T2. Abbreviations: BWC, body weight changes; ONS, oral nutritional supplementation.

**Table 1 nutrients-12-02516-t001:** Clinical characteristics of patients.

Variables	Total Sample T0 (*n* = 91), T1 (*n* = 70), T2 (*n* = 65)	Analyzed Sample T0 (*n* = 60), T1 (*n* = 56), T2 (*n* = 53) *n* (%) or Mean ± SD
Age (years)	60.6 ± 10.9	59.8 ± 10.1
Sex, male	70 (76.9)	53 (81.5)
Underweight patients by BMI T0 T1 T2	16 (18.0) 16 (23.9) 16 (25.0)	9 (13.8) 14 (23.3) 13 (22.0)
Normal weight patients by BMI T0 T1 T2	43 (48.3) 35 (52.2) 35 (54.7)	31 (47.7) 30 (50.0) 33 (55.9)
Overweight patients by BMI T0 T1 T2	30 (33.7) 16 (23.9) 13 (20.3)	25 (38.5) 16 (26.7) 13 (22.0)
Oral nutritional supplements use T0 T1 T2	32 (35.2) 47 (67.1) 47 (72.3)	18 (30.0) 35 (62.5) 36 (67.9)
Gastric or enteral tubes T0 T1 T2	10 (11.0) 14 (20.2) 11 (17.2)	4 (6.1) 13 (20.9) 10 (17.0)
Nutritional counseling		
T0 T1 T2	20 (22.0) 44 (63.8) 44 (67.7)	12 (18.5) 38 (61.3) 41 (68.3)
Tumor site Oral cavity ^a^ Nasal cavity Larynx Pharynx ^b^ Other ^c^	30 (33.0) 4 (4.4) 32 (35.2) 22 (24.2) 3 (3.3)	21 (32.3) 3 (4.6) 26 (40.0) 14 (21.5) 1 (1.5)
T Stage T1 T2 T3 T4 TX Not specified or unknown	8 (8.8) 21 (23.1) 28 (30.8) 28 (30.8) 4 (4.4) 2 (2.2)	8 (12.2) 17 (26.1) 21 (32.3) 17 (26.1) 2 (3.1)
N Stage N0 N1 N2 N3 NX Not specified or unknown	38 (41.8) 19 (20.9) 18 (19.8) 8 (8.8) 6 (6.6) 2 (2.2)	30 (46.2) 17 (26.2) 11 (16.9) 4 (6.2) 3 (4.6)
M Stage M0 M1 MX Not specified or unknown	55 (60.4) 4 (4.4) 26 (28.6) 6 (6.6)	40 (61.5) 2 (3.1) 20 (30.8) 3 (4.6)
Clinical Stage I II III IV Not specified or unknown	8 (8.8) 13 (14.3) 24 (26.4) 42 (46.2) 4 (4.4)	8 (12.3) 10 (15.4) 19 (29.2) 28 (43.1)
Mode of treatment Radiotherapy Surgery and radiotherapy Chemoradiotherapy Surgery and chemoradiotherapy Chemotherapy Surgery Other ^d^	21 (23.9) 10 (11.4) 37 (42.0) 7 (8.0) 1 (1.1) 5 (5.7) 7 (8.0)	16 (24.6) 10 (15.4) 33 (50.8) 6 (9.2)

Abbreviations: BMI, body mass index; SD, standard deviation; Treatment time points: T0, before or at beginning of treatment; T1, middle of treatment; T2, final of treatment. ^a^ tongue, mouth floor, and lip; ^b^ hypopharynx, oropharynx, and nasopharynx; ^c^ jaw, cervical and parathyroid; ^d^ Loss of follow-up before starting treatment.

**Table 2 nutrients-12-02516-t002:** Association of treatment time points and oral nutritional supplementation on energy, macro-, and micronutrient intake in head and neck cancer patients during (chemo) radiotherapy.

Dependent Variables Mean SE	Independent Variables—Treatment Time Points, Oral Nutritional Supplementation, and Treatment Time Points with Oral Nutritional Supplementation													
	T0 (*n* = 60)	T1 (*n* = 56)	T2 (*n* = 53)	*p*-Value	ONS 0/1 Time Overall (*n* = 82)	ONS 2/3 Times Overall (*n* = 87)	*p*-Value	T0		T1		T2		*p*-Value
								ONS 0/1 Time (*n* = 32)	ONS 2/3 Times (*n* = 28)	ONS 0/1 Time (*n* = 26)	ONS 2/3 Times (*n* = 30)	ONS 0/1 Time (*n* = 24)	ONS 2/3 Times (*n* = 29)	
Energy (kcal)	1829.55 ^a^ 85.12	1511.03 ^b^ 66.80	1804.08 ^a^ 82.33	**<0.001**	1707.20 73.31	1709.87 87.62	**0.976**	1859.03 100.05	1800.54 117.43	1505.06 81.79	1517.03 91.21	1778.34 96.57	1830.19 98.55	0.715
Energy (kcal/kg/day) ^d^	27.62 ^a^ 1.41	23.48 ^b^ 1.47	29.44 ^a^ 1.52	**<0.001**	25.97 1.30	27.51 1.93	0.422	27.25 1.47	28.00 2.21	22.61 1.73	24.38 1.97	28.42 1.72	30.49 2.20	0.829
Carbohydrate (g)	230.47 ^a^ 3.86	204.12 ^b^ 3.10	246.05 ^c^ 3.49	**<0.001**	224.81 3.44	227.62 3.51	0.553	226.18 4.98	234.83 5.69	205.89 4.27	202.37 4.00	243.97 5.34	248.15 4.80	0.296
Protein (g)	88.41 ^a^ 1.61	70.43 ^b^ 1.27	80.08 ^c^ 1.40	**<0.001**	77.80 1.25	80.83 1.59	0.123	88.42 2.14	88.41 2.42	69.36 1.78	71.51 1.68	76.79 1.77	83.52 2.20	0.147
Protein (g/kg/day) ^d^	1.34 ^a^ 0.06	1.10 ^b^ 0.05	1.31 ^a^ 0.06	**<0.001**	1.18 ^a^ 0.05	1.32 ^b^ 0.07	**0.031**	1.29 0.06	1.39 0.09	1.04 0.05	1.18 0.06	1.22 0.06	1.42 0.08	0.346
Lipids (g)	77.23 ^a^ 1.07	60.42 ^b^ 1.00	68.31 ^c^ 1.26	**<0.001**	68.88 1.08	67.75 1.18	0.498	79.23 1.40	75.28 1.62	59.89 1.35	60.96 1.37	68.86 2.01	67.76 1.78	0.176
Monounsaturated fat (g)	20.89 ^a^ 0.57	15.41 ^b^ 0.38	17.75 ^c^ 0.59	**<0.001**	18.37 0.49	17.40 0.54	0.169	21.72 0.71	20.10 0.76	15.47 0.54	15.36 0.57	18.45 0.78	17.07 0.85	0.417
Polyunsaturated fat (g)	21.33 ^a^ 0.86	15.43 ^b^ 0.54	18.02 ^c^ 0.70	**<0.001**	19.32 ^a^ 0.72	16.95 ^b^ 0.69	**0.005**	23.27 1.14	19.54 1.02	16.05 0.69	14.83 0.73	19.32 0.95	16.81 0.92	0.440
Saturated fat (g)	22.50 ^a^ 0.49	19.15 ^b^ 0.41	21.69 ^a^ 0.59	**<0.001**	20.44 0.43	21.71 0.52	0.061	22.43 0.58	22.57 0.71	18.52 0.56	19.80 0.61	20.55 0.95	22.90 0.75	0.178
Cholesterol (mg)	289.81 ^a^ 14.89	196.10 ^b^ 11.29	263.94 ^a^ 16.08	**<0.001**	247.68 12.59	245.57 15.85	0.906	291.92 17.82	287.72 21.51	196.82 14.40	195.37 14.51	264.43 19.31	263.45 23.98	0.995
Calcium (mg)	489.32 ^a^ 24.57	771.70 ^b^ 32.74	812.89 ^b^ 52.45	**<0.001**	562.76 ^a^ 25.13	808.59 ^b^ 50.56	**<0.001**	398.89 26.65	600.26 48.39	653.39 32.32	911.43 57.70	683.83 61.90	966.30 78.40	0.759
Iron (mg)	7.76 ^a^ 0.43	9.98 ^b^ 0.46	10.19 ^b^ 0.73	**<0.001**	8.33 ^a^ 0.32	10.25 ^b^ 0.85	**0.019**	7.46 0.40	8.07 0.71	8.75 0.51	11.39 1.00	8.87 0.72	11.71 1.29	0.148
Fiber (g)	20.31 ^a^ 0.82	17.86 ^b^ 0.86	19.80 ^a,b^ 0.65	**0.032**	20.00 0.75	18.62 0.82	0.174	21.56 1.14	19.14 1.05	18.01 1.17	17.72 1.18	20.60 0.80	19.03 0.96	0.582
Phosphorus (mg)	1007.34 ^a^ 24.66	1015.58 ^a^ 24.22	1127.24 ^b^ 23.77	**<0.001**	990.57 ^a^ 21.59	1110.16 ^b^ 30.58	**0.001**	960.26 27.12	1056.73 42.30	941.74 27.69	1095.21 34.85	1074.84 35.98	1182.20 33.48	0.373
Magnesium (mg)	212.44 ^a^ 8.74	252.40 ^b^ 9.94	251.03 ^b^ 9.69	**<0.001**	222.03 ^a^ 7.84	254.86 ^b^ 11.20	**0.005**	206.92 9.81	218.10 13.65	223.32 10.14	285.27 16.22	236.85 11.69	266.06 14.00	0.053
Manganese (mg)	1.80 ^a^ 0.08	2.15 ^b^ 0.09	2.42 ^b^ 0.11	**<0.001**	1.97 ^a^ 0.08	2.26 ^b^ 0.10	**0.011**	1.77 ^a^ 0.09	1.84 ^a,c^ 0.12	1.87 ^a,c^ 0.10	2.48 ^b^ 0.14	2.30 ^b,c^ 0.15	2.54 ^b^ 0.16	0.034
Niacin (mg)	15.35 0.57	15.78 0.58	16.50 0.47	0.120	15.03 ^a^ 0.43	16.75 ^b^ 0.63	**0.019**	15.15 0.70	15.55 0.88	14.08 0.79	17.68 0.88	15.92 0.56	17.10 0.72	0.067
Potassium (mg)	3648.17 ^a,b^ 100.35	3446.59 ^a^ 79.65	3718.62 ^b^ 97.27	**0.004**	3531.44 89.78	3675.18 90.79	0.161	3635.20 119.84	3661.19 137.94	3322.53 92.88	3575.28 122.48	3646.36 130.33	3792.31 112.15	0.460
Riboflavin (mg)	0.95 ^a^ 0.04	1.29 ^b^ 0.05	1.37 ^b^ 0.06	**<0.001**	1.01 ^a^ 0.04	1.39 ^b^ 0.07	**<0.001**	0.80 0.05	1.12 0.08	1.10 0.06	1.51 0.08	1.19 0.09	1.58 0.10	0.893
Sodium (mg)	3112.91 ^a^ 79.57	2393.95 ^b^ 68.84	2774.00 ^c^ 77.75	**<0.001**	2825.23 57.12	2666.08 85.65	0.052	3209.89 77.75	3018.87 113.46	2489.29 89.68	2302.26 94.11	2822.25 93.60	2726.58 98.14	0.734
Thiamine (mg)	1.01 ^a^ 0.04	1.21 ^b^ 0.05	1.37 ^c^ 0.06	**<0.001**	1.06 ^a^ 0.04	1.33 ^b^ 0.06	**<0.001**	0.93 0.04	1.09 0.06	1.03 0.06	1.43 0.08	1.25 0.06	1.50 0.09	0.098
Vitamin B6 (mg)	0.59 ^a^ 0.03	0.93 ^b^ 0.05	0.96 ^b^ 0.05	**<0.001**	0.68 ^a^ 0.04	0.96 ^b^ 0.06	**<0.001**	0.51 0.04	0.68 0.06	0.73 0.06	1.19 0.09	0.84 0.06	1.10 0.08	0.126
Vitamin C (mg)	77.85 5.91	96.64 6.54	91.48 7.76	0.054	74.67 ^a^ 4.32	104.40 ^b^ 9.20	**0.001**	63.93 5.61	94.79 11.32	82.27 7.26	113.53 11.04	79.15 9.88	105.73 12.43	0.894
Zinc (mg)	12.50 0.47	11.59 0.42	12.87 0.52	**0.022**	11.50 ^a^ 0.36	13.17 ^b^ 0.62	**0.008**	12.40 ^a,b^ 0.51	12.59 ^a,b^ 0.62	11.03 ^b^ 0.51	12.18 ^b^ 0.68	11.13 ^b^ 0.60	14.89 ^a^ 0.94	0.009

Abbreviations: ONS, oral nutritional supplementation; SE, standard error; Treatment time points: T0, before or at beginning of treatment; T1, middle of treatment; T2, final of treatment; Adjusted for intra-individual variability, proposed by [29]; Adjusted for total energy consumption by residual method, proposed by [30]. Generalized estimating equations model, adjusted: treatment, age, sex, tumor site, and stage. Bonferroni post-hoc test: Different superscript letters represent statistical difference (^a,b,c^) in pairwise comparisons, *p* < 0.05. Significant tests of model effects shown in bold. Overall ONS is the total patient data values representing the average of the three treatment time points (*n* = 169). ^d^ Reference for minimum recommended intake: 25 kcal/kg/day and 1 g protein/kg/day [33].

**Table 3 nutrients-12-02516-t003:** Percentage of macronutrient inadequacy, protein in g/kg/day, energy in kcal/kg/day, and mean values and standard deviation of cholesterol intake in head and neck cancer patients during (chemo) radiotherapy.

Energy and Nutrients	Percentage of Inadequacy *n* (%)								
	T0			T1			T2		
	Total (*n* = 60)	Without ONS (*n* = 42)	With ONS (*n* = 18)	Total (*n* = 56)	Without ONS (*n* = 21)	With ONS (*n* = 35)	Total (*n* = 53)	Without ONS (*n* = 17)	With ONS (*n* = 36)
Energy (>25 kcal/kg/day) ^b^	16 (26.7)	13 (31.0)	3 (16.7)	23 (43.4)	10 (50.0)	13 (39.4)	11 (21.2)	8 (47.1)	3 (8.6)
Carbohydrate (45–65%) ^a^	35 (58.3)	26 (61.9)	9 (50.0)	31 (55.4)	13 (61.9)	18 (51.4)	27 (50.9)	10 (58.8)	17 (47.2)
Protein (10–35%) ^a^	3 (5.0)	3 (7.1)	0 (0.0)	3 (5.4)	0 (0.0)	3 (8.6)	0 (0.0)	0 (0.0)	0 (0.0)
Protein (>1 g/kg/day) ^b^	7 (11.7)	6 (14.3)	1 (5.6)	16 (30.2)	8 (40.0)	8 (24.2)	4 (7.7)	4 (23.5)	0 (0.0)
Lipids– 20–35%) ^a^	32 (53.3)	23 (54.8)	9 (50.0)	31 (55.4)	11 (52.4)	20 (57.1)	20 (37.7)	10 (58.8)	10 (27.8)
Monounsaturated fat (15–20%) ^c^	57 (95.0)	39 (92.9)	18 (100.0)	56 (100.0)	21 (100.0)	35 (100.0)	52 (98.1)	16 (94.1)	36 (100.0)
Polyunsaturated fat (6–11%) ^c^	26 (43.3)	20 (47.6)	6 (33.3)	25 (44.6)	9 (42.9)	16 (45.7)	22 (41.5)	8 (47.1)	14 (38.9)
Saturated fat (< 10%) ^c^	38 (63.3)	26 (61.9)	12 (66.7)	35 (62.5)	12 (57.1)	23 (65.7)	29 (54.7)	10 (58.8)	19 (52.8)
Cholesterol (<300 mg) ^c^ Mean SD	294.1 90.3	282.4 74.2	321.6 117.9	199.3 71.7	200.6 54.8	198.5 81.0	269.9 117.6	270.8 107.4	269.5 123.5

Treatment time points: T0, before or at beginning of treatment; T1, middle of treatment; T2, final of treatment; ONS, oral nutritional supplementation: without or with use; **^a^** AMDR, acceptable macronutrient distribution range [32]; ^b^ Reference for minimum recommended intake: 25 kcal/kg/day and 1 g protein/kg/day [33]; ^c^ [34]. Cholesterol intake should be minimized while consuming a nutritionally adequate diet.

**Table 4 nutrients-12-02516-t004:** Prevalence of nutrient intake inadequacy using the estimated average requirement (EAR) method as the cut-off point, and comparison of intake with adequate intake (AI) in head and neck cancer patients during (chemo)radiotherapy.

Nutrients	Sex	Age Group (years)	DRI (EAR)	Prevalence of Inadequacy (%)								
				T0			T1			T2		
				Total (*n* = 60)	Without ONS (*n* = 42)	With ONS (*n* = 18)	Total (*n* = 56)	Without ONS (*n* = 21)	With ONS (*n* = 35)	Total (*n* = 53)	Without ONS (*n* = 17)	With ONS (*n* = 36)
Calcium (mg)	Male	31–70 >70	800 1000	62.17 77.94	79.67 93.19	40.13 53.98	30.50 46.41	51.20 77.04	21.48 34.83	28.43 42.07	75.17 93.32	14.23 25.46
	Female	31–50 >51	800 1000	62.17 77.94	79.67 93.19	40.13 53.98	30.50 46.41	51.20 77.04	21.48 34.83	28.43 42.07	75.17 93.32	14.23 25.46
Iron (mg)	Male	>31	6	21.77	23.27	17.36	17.11	18.14	14.01	15.62	27.09	9.51
	Female	31–50 >51	8.1 5	36.69 16.11	42.07 16.35	27.76 13.35	25.46 13.79	33.72 12.71	20.61 11.51	23.89 12.51	47.61 18.94	15.62 7.35
Phosphorus (mg)	Male	>31	580	7.49	6.68	7.08	6.18	5.26	6.18	2.87	10.56	0.52
	Female	>31	580	7.49	6.68	7.08	6.18	5.26	6.18	2.87	10.56	0.52
Magnesium (mg)	Male	>31	350	86.86	95.35	70.88	69.50	94.63	59.48	79.10	97.98	69.50
	Female	>31	265	61.41	71.90	47.61	43.25	65.91	35.94	43.64	77.94	28.77
Niacin (mg)	Male	>31	12	26.43	30.15	18.41	24.20	33.36	18.94	8.85	24.20	2.33
	Female	>31	11	21.48	24.83	14.46	20.05	28.43	15.15	5.26	17.62	1.02
Riboflavin (mg)	Male	>31	1.1	45.62	60.26	24.20	24.20	35.20	17.36	22.96	51.60	12.92
	Female	>31	0.9	33.00	43.25	16.11	16.11	24.83	10.93	16.60	34.83	8.69
Thiamine (mg)	Male	>31	1.0	42.47	48.01	30.85	29.12	38.21	24.20	16.35	36.32	6.94
	Female	>31	0.9	34.46	38.97	24.83	23.89	30.15	19.77	11.90	28.77	4.36
Vitamin B6 (mg)	Male	31–50 >51	1.1 1.4	84.61 96.08	96.86 99.83	63.31 80.78	49.60 66.28	81.33 95.73	39.36 54.78	49.20 73.57	87.29 98.50	35.94 62.55
	Female	31–50 >51	1.1 1.3	84.61 93.45	96.86 99.49	63.31 75.80	49.60 61.03	81.33 92.51	39.36 49.60	49.20 65.91	87.29 96.64	35.94 53.59
Vitamin C (mg)	Male	>31	75	39.74	49.60	27.43	26.43	37.07	20.90	32.64	50.00	27.09
	Female	>31	60	31.56	36.32	22.36	19.22	21.48	15.15	25.14	35.94	20.61
Zinc (mg)	Male	>31	9.4	12.71	14.92	7.64	26.11	30.15	24.20	21.19	42.47	11.51
	Female	>31	6.8	2.81	3.75	1.13	12.51	13.79	11.70	11.51	26.43	5.05
			**DRI (AI)**	**Consumption Comparison with AI-Below Recommendation *n* (%)**								
Fiber (g)	Male	31–50 >51	38 30	9(15.0) 36 (60.0)	6(14.3) 24(57.1)	3(16.7) 12(66.7)	9(16.1) 35(62.5)	4(19.0) 12(57.1)	5(14.3) 23(65.7)	8(15.1) 32(60.4)	3(17.6) 9(52.9)	5(13.9) 23(63.9)
	Female	31–50 >51	25 21	3(5.0) 6(10.0)	3(7.1) 4(9.5)	0(0.0) 2(11.1)	2(3.6) 6(10.7)	2(9.5) 1(4.8)	0(0.0) 5(14.3)	3(5.7) 7(13.2)	2(11.8) 2(11.8)	1(2.8) 5(13.9)
Manganese (mg)	Male	>31	2.3	41(68.3) ^a^	28(66.7)	13(72.2)	23(41.1) ^a^	10(47.6)	13(37.1)	16(30.2) ^a^	10(58.8)	6(16.7)
	Female	>31	1.8	6(10.0) ^a^	4(9.5)	2(11.1)	6(10.7) ^a^	4(19.0)	2(5.7)	6(11.3) ^a^	3(17.6)	3(8.3)
Potassium (mg)	Male	>31	3400	21(35.0) ^b^	13(31.0)	8(44.4)	20(35.7) ^b^	7(33.3)	13(37.1)	11(20.8) ^b^	8(47.1)	3(8.3)
	Female	>31	2600	3(5.0) ^b^	3(7.1)	0(0.0)	4(7.1) ^b^	2(9.5)	2(5.7)	1(1.9) ^b^	1(5.9)	0(0.0)
Sodium (mg)	Male	>31	1500	0(0.0) ^b^	0(0.0)	0(0.0)	4(7.1) ^b^	2(9.5)	2(5.7)	2(3.8) ^b^	1(5.9)	1(2.8)
	Female	>31	1500	0(0.0) ^b^	0(0.0)	0(0.0)	1(1.8) ^b^	0(0.0)	1(2.9)	0(0.0) ^b^	0(0.0)	0(0.0)

Treatment time points: T0, before or at beginning of treatment; T1, middle of treatment; T2, final of treatment; DRI, Dietary Reference Intake [32]; EAR, estimated average requirement; AI, adequate intake, ONS, oral nutritional supplementation: without or with use. ^a^ Below UL (tolerable upper intake level) of 11 mg. ^b^ UL not determined due to lack of toxicological indicator specific to excessive potassium and sodium intake.

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
