# Peer review of "Oral Nutritional Supplementation Affects the Dietary Intake and Body Weight of Head and Neck Cancer Patients during (Chemo) Radiotherapy"

_nutrients, 2020, doi:10.3390/nu12092516_

Round 1

Reviewer 1 Report

In this manuscript, Ferreira et al., studied Time of Treatment and Oral Nutritional Supplementation Affect the Dietary Intake and Body Weight of Head and Neck Cancer Patients during 4 (Chemo) radiotherapy.

Title words “Time of Treatment” is confusing, this should be defined properly. Alternatively, you may change your title. It is better to highlight your positive results.  “Early Oral Nutritional Supplementation Improves the Dietary Intake and Body Weight gain of Head and Neck Cancer Patients during (Chemo) radiotherapy”. This is a suggestion.

Details of oral nutritional supplementation (ONS) should be presented as a table or in supplementary data.

Page 4, line 47, “individuals who used ONS 2/3 times and those who used ONS 0/1 time”. Why the individuals used 0 and 1 time were put together?

Under 2.3. Demographic, clinical and nutritional assessment, and/or Figure 1.  Have you measured Waist circumference changes?

2.4. Dietary Assessment. Have you done any biochemical analysis for key micronutrients?

Page 14, line 34-35.  “we observed that those who used ONS showed less weight loss (except of the overweight) and consumed more protein (g/kg/day)”. Can you put a number here (xg/kg/day)?

Author Response

Uberlandia, Brazil

August 5th, 2020

Dear Editor and Reviewer#1,

Thank you for all valuable review regarding our manuscript nutrients-877690 entitled Oral Nutritional Supplementation Affects the Dietary Intake and Body Weight of Head and Neck Cancer Patients during (Chemo) radiotherapy”. We have taken into consideration all comments and suggestions made by the editor and reviewers, and modified our manuscript accordingly. We highlighted the changes in our manuscript in red and enumerated all points raised by the reviewers with a response for each one of them. The English language was reviewed by a proofreading service. Please, find enclosed the revised manuscript and the English review comprobatory certificate (page 5).

Yours sincerely,

Geórgia das Graças Pena

Graduate Program in Health Sciences

Federal University of Uberlandia,

Uberlandia, Minas Gerais, Brazil

Reviewer 2 Report

In this manuscript, the authors reported that oral nutritional supplementation (ONS) affects the body weight of head and neck cancer patients during chemotherapy and radiotherapy.

In the study, the authors concluded that nutritional counseling combined with prophylactic ONS should be initiated based on the diagnosis to optimize macronutrient intake and minimize weight loss so as to prevent worsening of the prognosis. If the authors did not find any association between the weight loss and the prognosis, how is it justified to report that initiating nutritional counseling prevents worsening of the prognosis?

Although ONS prevents weight loss and improves the nutritional status of head and neck cancer patients during chemotherapy and radiotherapy, most clinicians have already reported this phenomenon in clinical stages. If the authors cannot show the relationship between ONS and the prognosis, this study only replicates what is known in the field by the clinicians and lacks novelty.

Author Response

Uberlandia, Brazil

August 5th, 2020

Roy Zhang
Assistant Editor
Nutrients

Dear Editor and Reviewer#2,

Thank you for all valuable review regarding our manuscript nutrients-877690 entitled Oral Nutritional Supplementation Affects the Dietary Intake and Body Weight of Head and Neck Cancer Patients during (Chemo) radiotherapy”. We have taken into consideration all comments and suggestions made by the editor and reviewers, and modified our manuscript accordingly. We highlighted the changes in our manuscript in red and enumerated all points raised by the reviewers with a response for each one of them. The English language was reviewed by a proofreading service. Please, find enclosed the revised manuscript and the English review comprobatory certificate (page 5).

Yours sincerely,

Geórgia das Graças Pena

Graduate Program in Health Sciences

Federal University of Uberlandia,

Uberlandia, Minas Gerais, Brazil

Round 2

Reviewer 1 Report

I think this manuscript has been significantly improved and now warrants publication in Nutrients. 

Reviewer 2 Report

The manuscript has been revised well. I think this manuscript will be acceptable after some corrections have been done.